# TWIST1 Drives Cytotoxic CD8+ T-Cell Exhaustion through Transcriptional Activation of *CD274* (PD-L1) Expression in Breast Cancer Cells

**DOI:** 10.3390/cancers16111973

**Published:** 2024-05-22

**Authors:** Xiaobin Yu, Jianming Xu

**Affiliations:** 1Department of Molecular and Cellular Biology, Baylor College of Medicine, Houston, TX 77030, USA; xiaobiny@bcm.edu; 2Dan L Duncan Comprehensive Cancer Center, Baylor College of Medicine, Houston, TX 77030, USA

**Keywords:** breast cancer, TWIST1, EMT, metastasis, immunosuppression, PD-L1, immune-checkpoint-inhibitor therapy

## Abstract

**Simple Summary:**

TWIST1 is a transcription factor that induces epithelial-mesenchymal transition (EMT), and EMT is positively associated with programmed death ligand 1 (PD-L1) expression and immune escape in breast cancer cells. However, the relationship between TWIST1 and PD-L1 or immune evasion in breast cancer is unknown. In this study, we found that PD-L1 is a target gene of TWIST1. TWIST1 interacts with the TIP60 acetyltransferase complex in a BRD8-dependent manner to robustly upregulate PD-L1 expression. The knockdown of TWIST1 or BRD8 largely diminished PD-L1 expression and enhanced the CD8+ T-cell-mediated inhibition of breast cancer cell growth. Furthermore, the blockade of PD-L1 expressed in TWIST1-positive breast cancer cells also abolished the TWIST1-mediated suppression of CD8+ T cells and reinvigorated CD8+ T cells to inhibit cancer cells. These results suggest that TWIST1 may be a potential target for silencing PD-L1 expression to prevent the exhaustion of cytotoxic CD8+ T cells.

**Abstract:**

In breast cancer, epithelial-mesenchymal transition (EMT) is positively associated with programmed death ligand 1 (PD-L1) expression and immune escape, and TWIST1 silences ERα expression and induces EMT and cancer metastasis. However, how TWIST1 regulates PD-L1 and immune evasion is unknown. This study analyzed TWIST1 and PD-L1 expression in breast cancers, investigated the mechanism for TWIST1 to regulate PD-L1 transcription, and assessed the effects of TWIST1 and PD-L1 in cancer cells on cytotoxic CD8+ T cells. Interestingly, TWIST1 expression is correlated with high-level PD-L1 expression in ERα-negative breast cancer cells. The overexpression and knockdown of TWIST1 robustly upregulate and downregulate PD-L1 expression, respectively. TWIST1 binds to the PD-L1 promoter and recruits the TIP60 acetyltransferase complex in a BRD8-dependent manner to transcriptionally activate PD-L1 expression, which significantly accelerates the exhaustion and death of the cytotoxic CD8+ T cells. Accordingly, knockdown of TWIST1 or BRD8 or inhibition of PD-L1 significantly enhances the tumor antigen-specific CD8+ T cells to suppress the growth of breast cancer cells. These results demonstrate that TWIST1 directly induces PD-L1 expression in ERα-negative breast cancer cells to promote immune evasion. Targeting TWIST1, BRD8, and/or PD-L1 in ERα-negative breast cancer cells with TWIST1 expression may sensitize CD8+ T-cell-mediated immunotherapy.

## 1. Introduction

Genome instability is a hallmark of cancer, which results in the accumulation of somatic gene mutations, leading to the production of cancer cell-specific neoantigens. Presumably, these neoantigens presented by cancer cells should be detected by the immune surveillance system, and these cancer cells should be eliminated by the responses of the immune system. However, many cancers actually escape from immune suppression in multiple ways. Firstly, tumor-associated macrophages, infiltrated myeloid-derived suppressor cells, and regulatory T cells may contribute to the formation of a suppressive tumor microenvironment that prevents cancer cells from immune destruction [1]. Secondly, although tumor antigen-specific cytotoxic CD8+ T cells can kill cancer cells [2], the prolonged exposure of these CD8+ T cells to cancer cells leads to their exhaustion, causing the CD8+ T cells to lose their cytotoxicity to cancer cells [3,4]. Thirdly, cancer cells can express high-level immune checkpoint proteins, such as the programmed death ligand 1 (PD-L1) protein coded by the *CD274* gene, to protect themselves from being attacked by the cytotoxic effector T cells [5]. PD-L1 expressed in cancer cells interacts with its receptor, the programmed death protein 1 (PD-1), on immune cells to suppress immune cell function [6]. The overexpression of PD-L1 in cancer cells is correlated with the poor prognosis of patients with advanced non-small cell lung cancer (NSCLC) [7], breast cancer [8], head/neck cancer [9], gastric cancer [10], and prostate cancer [11]. PD-L1-neutralizing antibodies, including atezolizumab and avelumab, and PD-1-neutralizing antibodies, including nivolumab, pembrolizumab, and dostarlimab, have been used to treat multiple types of cancers such as melanoma, renal cell carcinoma, squamous lung cancer, and metastatic NSCLC [12,13,14]. Combination immunotherapy using PD-L1/PD-1 immune checkpoint inhibitors (ICIs) and chemotherapy is moderately effective and has been clinically approved for treating triple-negative breast cancer (TNBC) [15,16,17]. Currently, one of the major challenges for breast cancer immunotherapy using ICIs is identifying reliable predictive biomarkers of response for patient selection and treatment optimization. Current markers, such as PD-L1 expression and tumor mutational burden, are still inconsistent for predicting outcomes of ICI therapies across breast cancer subtypes.

TWIST1 is a transcription factor that induces epithelial-mesenchymal transition (EMT) to promote cancer cell invasion, metastasis, stemness, and drug resistance when it is expressed in breast and other cancer cells [18,19,20,21,22,23,24]. We have previously shown that the expression of TWIST1 reprogrammed the non-metastatic, estrogen receptor α-positive (ER+) and HER2-negative (HER2−) MCF7 breast cancer cells with high-level expression of luminal epithelial genes into metastatic, ER-negative (ER−) and HER2-negative (HER2−) breast cancer cells with a more triple-negative breast cancer (TNBC)-like phenotype, as evidenced by their high-level expression of mesenchymal genes and very low level of epithelial genes [25]. Conversely, the knockdown of TWIST1 in ER−/HER2− breast cancer cells such as BT549 and 4T1 cells with endogenous TWIST1 expression significantly inhibited their invasion and metastasis capabilities [25,26,27,28]. We also have reported that only a small subset of primary mouse breast tumor cells but almost all circulating tumor cells express Twist1 and exhibit partial EMT, while the knockout of *Twist1* can prevent these partial EMT and intravasation phenotypes from occurring in these tumor cells [20]. In addition, TWIST1 also induces therapeutic resistance by suppressing the expression of estrogen receptor α (ERα), which serves as the target of antiestrogen by inhibiting p53 and upregulating AKT2 [25,28,29,30,31].

Regarding the molecular mechanisms for TWIST1-mediated transcription in breast cancer cells, we and others have demonstrated that TWIST1 interacts with and recruits the nucleosome remodeling and deacetylase (NuRD) complex to repress the expression of epithelial genes such as *CDH1* (E-cadherin), *ESR1* (ERα), and *FOXA1*, while it interacts with and recruits the NuA4/TIP60 histone acetyltransferase (HAT) complex (TIP60-Com) to activate the expression of mesenchymal genes such as *SNAI2* and growth genes such as *MYC*, *WNT5A*, and *CCNE1* [26,32,33]. The TIP60-Com contains at least 16 subunits, including BRD8, p400, EPC1, and the TIP60 HAT [34,35]. The major acetylation targets of TIP60 are lysine residues (Ks) on nucleosomal histone H4, histone H2AK5, histone variants H2AZ and H2AX, and non-histone proteins such as p53 and TWIST1 [32,36]. On the chromatin, TIP60-Com functions to activate transcription involved in the regulation of DNA repair, cell homeostasis, stress response, stem cell renewal, and EMT [26,37,38,39,40,41]. BRD8 is a bromodomain (BD)-containing protein with two BDs. BD binds acetyl-Lys (acK) motifs in histones and non-histone proteins [42]. We have shown that TWIST1 interacts with TIP60-Com in a BRD8-dependent manner to activate mesenchymal gene expression in ER- breast cancer or TNBC cells [26].

Interestingly, EMT has been positively associated with cancer immune escape, though the molecular mechanisms are largely unclear [43]. Among the EMT-inducing transcription factors, SNAI1 and SNAI2 have been suggested to promote PD-L1 expression through upregulating ZEB1. ZEB1 has been shown to indirectly upregulate PD-L1 expression through repressing the expression of miR-200, a negative regulator of the PD-L1 mRNA [44,45,46,47,48]. However, how TWIST1, a master EMT-inducing transcription factor, regulates PD-L1 expression and breast cancer immune escape is still unknown. In this study, we found that high levels of PD-L1 expression are associated with TWIST1 expression in ER−/HER2− breast tumors and TNBC cell lines. TWIST1 binds the *CD274* promoter and recruits TIP60-Com to the promoter to robustly upregulate PD-L1 expression, which accelerates the exhaustion and death of the tumor antigen-specific CD8+ effector T cells and prevents the TWIST1-expressing breast cancer cells from been killed by these CD8+ T cells.

## 2. Materials and Methods

### 2.1. Data Mining of Human Breast Cancer Transcriptomes

Publicly accessible METABRIC datasets contain transcriptomes of a large number of human breast tumors [49,50]. Using their normalized expression data available from the cBioPortal “https://www.cbioportal.org (accessed on 17 May 2024)” [51], we obtained and compared the mRNA expression profiles of *TWIST1*, *VIM*, *SNAI2*, *CDH1*, *ESR1* and *CD274* in different subtypes of breast tumors. Tukey’s multiple comparison tests were performed to compare the median log expression of each gene for each molecular subtype of breast cancer. A *p*-value < 0.05 was considered statistically significant. The RNA-Seq datasets of the Metastatic Breast Cancer Project [52] were downloaded from the cBioPortal for Cancer Genomics database “https://www.cbioportal.org (accessed on 17 May 2024)”. The metastatic breast cancer datasets were divided into TWIST1-high and TWIST1-low subgroups, and the expression levels of *CD274* mRNA were compared between these two subgroups. The correlation between *TWIST1* and *CD274* mRNA expression levels in the entire metastatic breast cancer cohort was determined by the Pearson test, where *p* < 0.05 is considered significant.

### 2.2. Western Blot

Western blot was performed as described previously [26]. Cells were lysed with RIPA buffer containing 10 mM Tris-HCl, pH 8.0, 1 mM EDTA, 0.5 mM EGTA, 1% Triton X-100, 0.1% sodium deoxycholate, 0.1% SDS, and 140 mM NaCl. Protein concentration was determined by the BCA protein assay reagent (ThermoFisher Scientific, Waltham, MA, USA). Then, 5 μg of total protein was separated by SDS-PAGE and blotted onto a nitrocellulose membrane. The membrane was incubated with primary antibodies and matched secondary antibodies conjugated with horseradish peroxidase (HRP). The activity of the HRP-conjugated secondary antibody (Bio-rad, Hercules, CA, USA) was visualized using the SuperSignal West Pico Chemiluminescent Substrate (Thermo Scientific, Waltham, MA, USA). Primary antibodies included those against TWIST1 (sc-81417, Santa Cruz, Dallas, TX, USA), Ovalbumin (sc-65984, Santa Cruz), GAPDH (sc-32233, Santa Cruz), PD-L1 (13684S, Cell Signaling Technology, Danvers, MA, USA), BRD8 (ab17969, Abcam, Cambridge, UK), and ACTIN (A5441, Millipore Sigma, Burlington, MA, USA).

### 2.3. Cell Culture and Generation of Cell Lines

MCF7 cells were obtained from ATCC and cultured in DMEM (Fisher Scientific, Hampton, NH, USA) supplemented with 10% fetal bovine serum (FBS), 4.5 g/L glucose, 0.36 mg/mL L-glutamine, 100 I.U./mL penicillin and 100 μg/mL streptomycin at 37 °C with 5% CO_2_. MCF7-Ctrl1 and MCF7-TWIST1 cells were generated as described previously [26]. Briefly, 300,000 MCF7 cells in 60 mm dishes were transfected using 3.75 μL of Lipofectamine 3000 (Life technologies, Waltham, MA, USA) and 5 μg of empty pcDNA3.1 plasmid for generating MCF7-Ctrl1 cell lines or 5 μg of pcDNA3.1-Flag-TWIST1 expression plasmid for generating MCF7-TWIST1 cell lines. Individual colonies surviving in the growth-selection medium containing 0.5 mg/mL G418 (Sigma-Aldrich, St. Louis, MO, USA) were isolated, expanded, and assayed by Western blot for TWIST1 expression. MCF7-Ctrl1-2b and MCF7-TWIST1-Ctrl2b cell lines were generated by transducing MCF7-Ctrl1 and MCF7-TWIST1 cells, respectively, with the pGIPZ lentivirus that expresses a non-targeting control shRNA and the puromycin-resistant marker as described previously [26]. MCF7-Ctrl1-BRD8KD and MCF7-TWIST1-BRD8KD cell lines were generated as described previously by transducing MCF7-TWIST1 cells with the pGIPZ lentivirus that expresses a BRD8 mRNA-targeting shRNA and the puromycin-resistant marker [26]. To make MCF7-Ctrl1-2o, MCF7-TWIST1-Ctrl2o, MCF7-Ctrl1-OVA and MCF7-TWIST1-OVA cell lines, lentivirus particles for expressing mCherry or mCherry and ovalbumin (OVA) were packaged in 293T cells, as described previously [26], and used to transduce MCF7-Ctrl1 and MCF7-TWIST1 cells. The cells that stably expressed mCherry were isolated by flow cytometry, expanded, and examined by Western blot for OVA expression.

BT549 cells were obtained from ATCC and cultured in RPMI-1640 medium with 10% FBS, 5 μg/mL insulin, 100 I.U./mL penicillin, and 100 μg/mL streptomycin at 37 °C with 5% CO_2_. The BT549-Ctrl1, BT549-TWIST1KD, BT549-Ctrl1-2b, BT549-Ctrl1-BRD8KD, BT549-Ctrl1-2o, BT549-Ctrl1-OVA, BT549-TWIST1KD-Ctrl2o, and BT549-TWIST1KD-OVA cell lines were generated using similar reagents and methods to those used for generating MCF7-derived cell lines as described above or previously [26]. The relationships and selection markers of all MCF7 cell- and BT549 cell-derived cell lines used in this study are summarized in Table 1.

### 2.4. RT-qPCR

Total RNA was isolated using TRIzol reagent (Life technologies, Grand Island, NY, USA) from different cell lines in culture. Reverse transcription was performed with 2 μg RNA by using the High-capacity cDNA Reverse Transcription kit (Applied Biosystems, Waltham, MA, USA). Quantitative real-time PCR (qPCR) for measuring *CD274* mRNA was performed using gene-specific primer pairs (Forward: 5′-GGCCCAAGCACTGAAAATGG-3′, and Reverse: 5′-CAGGCTCCCTGTTTGACTCC-3′), and matched Universal TaqMan probes (Roche, Nutley, NJ, USA). The relative mRNA expression level was normalized to the level of the endogenous ACTB (β-actin) mRNA.

### 2.5. ChIP-qPCR Assay

ChIP assays were performed by following the protocol of SimpleChIP Enzymatic Chromatin IP Kit (Cell Signaling, Danvers, MA, USA). Briefly, 2 × 10^7^ of MCF7 cell-derived and BT549 cell-derived cell lines on culture plates were cross-linked with 1% formaldehyde for 10 min at room temperature and quenched with 0.125 M Glycine. Fixed cells were washed and lysed in a buffer containing 10 mM EDTA, 50 mM Tris-HCl, pH 8.0, 0.33% SDS, 0.5% Empigen BB, and protease inhibitor cocktails. The cross-linked protein–DNA complexes in cell lysates were digested with micrococcal nuclease and sonicated using a Branson Sonifier 250 to shear genomic DNA to a length range of 150–500 bp. Sheared samples were pre-cleaned and then immunoprecipitated with 2 μg antibody overnight at 4 °C with rotation, followed by incubation with protein A/G beads for 4 h at 4 °C. Immunoprecipitated DNA was eluted with 120 μL elution buffer containing 2 mM EDTA, 100 mM NaCl, 20 mM Tris-HCl, pH 8.0, 0.5% Triton X-100, 0.1 M NaHCO_3_ and 1% SDS. The eluted DNA was extracted with phenol-chloroform, recovered by precipitation, and measured by TaqMan-qPCR using a 5′ forward primer (5′-GTAGGGAGCGTTGTTCCTCC), a 3′ reverse primer (5′-GCACTTTAGGACGGAGGGTC), and a matched Universal TaqMan probe (Roche, Nutley, NJ, USA) in the promoter region of the *CD274* gene. As a negative control, an equal amount of ChIP-grade non-immune mouse IgG was used to replace the specific antibody. qPCR analysis of a gene desert region in chromosome 2 using a forward primer, 5′-CATCCCTGGACTGATTGTCA, and a reverse primer, 5′-GGTTGGCCAGGTACATGTTT, served as a negative control for the non-specific immunoprecipitation of protein–DNA complexes [53].

### 2.6. Isolation of OT-1 Mouse CD8+ T Cells

The OT-1 transgenic mouse line (#003831, The Jackson Lab, Bar Harbor, ME, USA) expresses a T-cell receptor on the CD8+ T cells that specifically recognizes amino acid residues 257–264 of the ovalbumin presented by MHC-I of the antigen-presenting cells [54]. To isolate OT-1 CD8+ T cells, OT-1 mice were euthanized, and their spleens were isolated. The spleens were put in a 70 μM filter strainer (352350, Corning, Corning, NY, USA) sitting on a 50 mL Falcon tube, smashed with a 5 mL syringe plunger, and flushed with phosphate-buffered saline (PBS) containing 0.5% bovine serum albumin (BSA) and 2 mM EDTA. The filtered cell suspension was centrifuged at 1500 rpm at room temperature for 5 min and washed with the same buffer. CD8+ T cells were prepared using a mouse CD8 T-cell isolation kit (130-104-075, Miltenyi Biotec, Auburn, CA, USA). The red blood cells (RBCs) in the enriched CD8+ T-cell preparation by the kit were removed by incubating the cell suspension with 1X RBC lysis buffer (00-4300-54, Thermo Fisher, Waltham, MA, USA) for 5 min. The cells were collected by centrifuge, re-suspended in DMEM containing 10% FBS, 100 I.U./mL penicillin and 100 μg/mL streptomycin, and activated and expanded by treating the cells with the Dynabeads Mouse T-activator CD3/CD28 reagent (11452D, Thermo Fisher) for 4 days.

### 2.7. Breast Cancer Cell Proliferation Assay

MCF7-derived or BT549-derived control, overexpression, or knockdown cell lines were suspended in DMEM supplemented with 10% fetal bovine serum (FBS), 4.5 g/liter glucose, 0.36 mg/mL L-glutamine, 100 I.U./mL penicillin and 100 μg/mL streptomycin and seeded into a 96-well tissue culture plate at a density of 2500 cells per well. Then, 10,000 CD8+ T cells from OT-1 mice were added to each well and cocultured with MCF7- or BT549-derived breast cancer cells in the presence or absence of 0.5 μg/mL of anti-PD-L1 monoclonal antibody (mAb) (13684S, Cell Signaling Technology, Danvers, MA, USA) at 37 °C for 3 days. For measuring the viability of adherent breast cancer cells, the non-adherent T cells were collected with the culture medium from each well. Then, 100 μL of fresh culture medium and 20 μL of CellTiter 96 Aqueous One Solution (Promega, Madison, WI, USA) were sequentially added to each well. Cells in the plate were incubated at 37 °C for 3 h. The absorbance was measured at 490 nm using a Synergy HT plate reader (BioTek, Winooski, VT, USA).

### 2.8. Vitality and Apoptosis Assays of CD8+ T Cells

The OT-1 CD8+ T cells collected from the coculture described above were washed with ice-cold PBS, re-suspended in 98 μL of PBS containing 2% FBS and 1 mM EDTA, and mixed with 2 μL of PD-1 antibody for flow cytometry (130-111-952, Miltenyi Biotec). The cells in the mixture were incubated for 30 min at 4 °C in the dark and then diluted by adding 150 μL buffer. The number of PD-1-high CD8+ T cells was determined by flow cytometry. In addition, apoptotic CD8+ T cells were determined using a BD Pharmingen^TM^ FITC Annexin V Apoptosis Detection kit I (556547, BD Biosciences, Franklin Lakes, NJ, USA).

### 2.9. Statistical Analyses

For all experiments, data were collected from at least three biological replicates to ensure adequate power (>80%), except those indicated. Data were expressed as mean ± standard deviation (SD). Graphpad Prism 10 Software (GraphPad, La Jolla, CA, USA) was used to perform an unpaired two-sided Student’s *t*-test to analyze the difference between two datasets. A one-way ANOVA test was used to analyze the differences among three or more datasets. Pearson analysis was used to determine the correlation relationship between *TWIST1* and *CD274* mRNA expression levels. In all statistical analyses, a *p*-value < 0.05 was considered statistically significant.

## 3. Results

### 3.1. TWIST1 Robustly Upregulates PD-L1 Expression in Breast Cancer Cells

We previously reported that TWIST1 induces the transcriptional activation of mesenchymal genes such as *VIM* (Vimentin) and *SNAI2* (Snail 2) but induces the transcriptional repression of epithelial genes such as *CDH1* (E-cadherin) and *ESR1* (ERα) in breast cancer cells (Figure 1A). In particular, the expression of TWIST1 in MCF7 breast cancer cells induced *Vim* and *Snai2* expression but repressed *CDH1* and *ESR1* expression. Consequently, this ectopic expression of TWIST1 is sufficient to reprogram MCF7 cells, which are non-metastatic luminal epithelial ER+/HER2− breast cancer cells, into metastatic basal-like ER−/HER2− breast cancer cells with mesenchymal gene expression [27,28,55,56]. Interestingly, the analysis of our transcriptomic data [26] also revealed a robust increase in *CD274* mRNA in TWIST1-expressing MCF7 (MCF7-TWIST1) cells versus MCF7 control (MCF7-Ctrl1) cells harboring an empty expression vector (Figure 1B, Table 1). To assess the correlation of TWIST1 expression with its regulated genes in human breast tumors, we analyzed the METABRIC transcriptomic datasets downloaded from cBioPortal [49,50,51]. Among five subtypes of human breast tumors, the expression levels of *TWIST1* and its upregulated genes, *Vim* and *SNAI2*, are the highest, while its repressed target genes, *CDH2* and *ESR1*, are the lowest in ER−/HER2− breast cancers (Figure 1C). The expression level of *CD274* mRNA is also the highest in ER−/HER2− breast tumors (Figure 1D, left panel). Because TWIST1 expression in breast tumor cells is associated with metastasis [20], we further analyzed the expression relationship between *TWIST1* and *CD274* mRNAs in another dataset obtained from metastatic breast cancer samples [52]. We found that the average expression level of CD274 mRNA in the TWIST1-high subgroup is significantly higher than that in the TWIST1-low subgroup. Pearson analysis also revealed a significant positive correlation relationship between *TWIST1* and *CD274* mRNA expression levels in individual tumor samples of this dataset (Figure 1D, central and right panels). Furthermore, PD-L1 (CD274) protein is expressed at low levels in ER+/HER2−/TWIST1− epithelial breast cancer cell lines, including MCF7, ZR-75-1, and T47D, while it is at high levels in ER−/HER2−/TWIST1+ TNBC cell lines, including MDA-MB-436, BT549 and SUM1315 (Figure 1E). Collectively, these results suggest that the expression levels of *TWIST1* and *CD274* (PD-L1) are positively correlated in ER−/HER2−/TWIST1+ breast cancer cells, including the TWIST1-reprogrammed MCF7-TWIST1 cell lines.

To establish the regulatory relationship between TWIST1 and *CD274*, we assayed the effects of TWIST1 overexpression in MCF7-TWIST1 cells and TWIST1 knockdown (KD) in BT549-TWIST1KD cells (Table 1). We found that the overexpression of TWIST1 in MCF7-TWIST1 cells robustly upregulated *CD274* mRNA and its protein PD-L1 (Figure 2A,B), while the knockdown of TWIST1 in BT549-TWIST1KD cells drastically reduced *CD274* mRNA and PD-L1 protein (Figure 2C,D). These results demonstrate that TWIST1 is required for a high expression of PD-L1 in these ER−/HER2−/TWIST1+ breast cancer cells.

### 3.2. TWIST1 Binds the CD274 Promoter and Recruits TIP60-Com in a BRD8-Dependent Manner

The human *CD274* gene is located at the region from bp 5,450,542 to bp 5,470,554 in chromosome 9, and its promoter is located within a 500 bp sequence adjacent to its transcriptional start site (TSS), as symbolled by the peaks of acetylated histones H3K9 and H3K27 in MCF7 and MDA-MB-468 cells (Figure 3A). Interestingly, our ChIP-Seq analysis detected a sharp TWIST1-binding peak that overlapped with the *CD274* promoter in MCF7-TWIST1 cells, the specificity of which is supported by using MCF7-Ctrl1 cells without TWIST1 expression as a negative control (Figure 3A). ChIP-qPCR analysis further confirmed that TWIST1 is specifically associated with the *CD274* promoter in both MCF7-TWIST1 cells with ectopic TWIST1 expression and BT549-Ctrl1 cells with endogenous TWIST1 expression (Table 1, Figure 3B,C). These results indicate that *CD274* is a direct target gene of TWIST1.

We previously reported that TWIST1 interacts with TIP60-Com, a histone acetyltransferase complex containing BRD8, TIP60, P400, EPC1 and other components, in a BRD8-dependent manner to acetylate histone H4 for the transcriptional activation of its target genes, such as *SNAI2* and *MYC* [26]. To determine whether TWIST1 also recruits this complex to the *CD274* promoter, we performed ChIP-qPCR assays to assess the occupancy statuses of multiple components of TIP60-Com. We first used a gene desert region known to have no transcription factor binding as a negative-control chromatin region for antibodies, and we detected no occupancy of all examined proteins, including TWIST1, BRD8, TIP60, P400, EPC1 and acetylated histone H4 (acH4) in this region in both MCF7-TWIST1 and BT549-Ctrl1 cells with TWIST1 expression (Table 1, Figure 3B,C). We then performed ChIP-qPCR assays for the *CD274* promoter region using the antibodies to the same proteins. In both MCF7-TWIST1 and BT549-Ctrl1 cells, we detected abundant associations of the components of TIP60-Com, including BRD8, TIP60, P400, and EPC1, in addition to TWIST1, and high levels of acH4 with the *CD274* promoter. However, in TWIST1-negative MCF7-Ctrl1 cells and TWIST1-knockdown BT549-TWIST1KD cells (Table 1), no significant occupancies of the components of TIP60-Com or TWIST1 were detected (Figure 3B,C). Importantly, the knockdown of BRD8 in either MCF7-TWIST1-BRD8KD or BT549-Ctrl1-BRD8KD cells (Table 1) did not affect TWIST1 binding to the *CD274* promoter, but it diminished the recruitments of TIP60-Com components to the *CD274* promoter (Figure 3B,C). Collectively, these results indicate that TWIST1 effectively recruits TIP60-Com to the *CD274* promoter in a BRD8-dependent manner.

### 3.3. TWIST1 Requires TIP60-Com to Transcriptionally Activate CD274 Expression

Since BRD8 is required for TWIST1 to recruit TIP60-Com to the *CD274* promoter, we assessed the role of TIP60-Com in TWIST1-mediated transcriptional activation of the *CD274* gene by knocking down BRD8. We found that in TWIST1-negative MCF7-Ctrl1-2b cells and TWIST1-knockdown BT549-TWIST1KD-Ctrl2b cells (Table 1), both *CD274* mRNA and its protein PD-L1 are expressed at low levels, suggesting that this low-level *CD274* expression is independent of TWIST1. The knockdown of BRD8 in MCF7-Ctrl1-BRD8KD cells without TWIST1 expression and BT549-TWIST1KD-BRD8KD cells with residual TWIST1 expression (see Table 1 for cell lines) had no significant effects on the low-level *CD274* mRNA and its protein PD-L1 expression, suggesting that BRD8 does not regulate *CD274* expression in the absence of TWIST1. Importantly, *CD274* and PD-L1 are expressed at high levels in MCF7-TWIST1-Ctrl2b and BT549-Ctrl1-2b cells (Table 1) with high-level TWIST1 expression, and the knockdown of BRD8 in MCF7-TWIST1-BRD8KD and BT549-Ctrl1-BRD8KD cells with high-level TWIST1 expression (Table 1) dramatically reduced *CD274* and PD-L1 expression (Figure 4). These results, together with the results shown in Figure 3, demonstrate that TWIST1 relies on BRD8 to recruit TIP60-Com to the *CD274* promoter to activate its mRNA and protein expression.

### 3.4. TWIST1 Expressed in Breast Cancer Cells Promotes Their Immune Evasion in Culture by Accelerating CD8+ T-Cell Exhaustion and Death

PD-L1 on the surface of cancer cells binds its receptor PD-1 to the surface of cytotoxic effector T cells such as CD8+ T cells to inhibit T-cell activation and promote T-cell exhaustion and death, resulting in the immune escape of these cancer cells [59]. To examine whether TWIST1-upregulated *CD274*/PD-L1 expression in breast cancer cells promotes immune evasion, we established a coculture system consisting of breast cancer cells with the expression of ovalbumin (OVA), a tumor antigen mimic, and CD8+ T cells isolated from OT-1 mice that express an OVA-specific T-cell receptor [54]. We generated MCF7-Ctrl1-OVA and MCF7-TWIST1-OVA cell lines through the lentivirus-mediated expression of ovalbumin and the mCherry cell-sorting marker in MCF7-Ctrl1 and MCF7-TWIST1 cell lines, respectively. We also produced BT549-Ctrl1-OVA and BT549-TWIST1KD-OVA cell lines using the same approach as with BT549-Ctrl1 and BT549-TWIST1KD cell lines, respectively (Table 1, Figure 5). In the coculture, OT-1 CD8+ T cells inhibited the growth of the TWIST1−/OVA+ MCF7-Ctrl1-OVA cells by 20% versus the TWIST1−/OVA− MCF7-Ctrl1-2o cells, while they only inhibited the growth of the TWIST1+/OVA+ MCF7-TWIST1-OVA cells by 11% versus the TWIST1+/OVA− MCF7-TWIST1-Ctrl2o cells (Table 1, Figure 6A). Similarly, OT-1 CD8+ T cells inhibited the growth of the TWIST1-low/OVA+ BT549-TWIST1KD-OVA cells by 14% versus the TWIST1-low/OVA− BT549-TWIST1KD-Ctrl2o cells, but they only inhibited the growth of the TWIST1-high/OVA+ BT549-Ctrl1-OVA cells by 8% versus the TWIST1/OVA− BT549-Ctrl1-2o cells (Table 1, Figure 6B). These results suggest that TWIST1 expressed in breast cancer cells drives immune evasion from CD8+ T-cell-mediated immunity in vitro.

To examine the effect of TWIST1-upregulated PD-L1 in breast cancer cells on CD8+ T-cell exhaustion, we measured the percentage of OT-1 CD8+ T cells with high PD-1 (CD279) expression, which is a marker of CD8+ T-cell exhaustion [60]. The percentages of high PD-1 CD8+ T cells in the coculture of MCF7-TWIST1-Ctrl2o and MCF7-TWIST1-OVA cells with high-level TWIST1 and PD-L1 expression were significantly higher than in the coculture of MCF7-Ctrl1-2o and MCF7-Ctrl1-OVA cells without TWIST1 expression (Figure 6C). In agreement with this finding, the percentage of high PD-1 CD8+ T cells was also much higher in coculture with BT549-Ctrl1-2o and BT549-Ctrl1-OVA cells with high TWIST1 expression versus BT549-TWIST1KD-Ctrl2o and BT549-TWIST1KD-OVA cells with low TWIST1 expression (Figure 6D). We further stained CD8+ T cells in the cocultures with propidium iodide (PI) and measured the percentages of PI-positive dead CD8+ T cells by flow cytometry. We found that the percentage of PI-positive dead CD8+ T cells in the cocultures with MCF7-TWIST1-Ctrl2o and MCF7-TWIST1-OVA cells was higher than that in the cocultures with MCF7-Ctrl1-2o and MCF7-Ctrl1-OVA cells (Figure 6E). The percentages of PI-positive CD8+ T cells in the cocultures with BT549-Ctrl1-2o and BT549-Ctrl1-OVA cells were also higher than that in the cocultures with BT549-TWIST1KD-Ctrl2o and BT549-TWIST1KD-OVA cells (Figure 6F). Collectively, these results indicate that breast cancer cells with a TWIST1-promoted high expression of PD-1 significantly accelerate CD8+ cytotoxic T-cell exhaustion and death.

### 3.5. Inhibition of TWIST1-Upregulated PD-L1 (CD274) in Breast Cancer Cells Strongly Sensitizes CD8+ T Cells to Kill Cancer Cells

To test whether the blockade of the TWIST1-upregulated PD-L1 in breast cancer cells to bind to PD-1 in the CD8+ T cells can enhance the function of CD8+ T cells to kill the breast cancer cells, we treated the cocultures with a PD-L1-neutralizing antibody. Our assays revealed that this antibody treatment robustly enhanced the growth inhibition of MCF7-TWIST1-OVA cells with high TWIST1 and PD-L1 expression but exerted no obvious effect on MCF7-Ctrl1-OVA cells with no TWIST1 and low PD-L1 expression. Consistent results were also obtained from treated BT549-Ctrl1-OVA with high TWIST1 and PD-L1 expression and BT549-TWIST1KD-OVA cells with low TWIST1 and PD-L1 expression (Figure 7A,B). Accordingly, PD-L1 antibody treatment also rescued CD8+ T cells from exhaustion in the cocultures with MCF7-TWIST1-OVA cells or with BT549-Ctrl1-OVA cells, as evidenced by the significant reduction in the number of high-PD-1 CD8+ T cells (Figure 7C,D). These results demonstrate that PD-L1 antibody treatment can effectively overcome the immune evasion of breast cancer cells promoted by TWIST1-upregulated PD-L1, suggesting that TWIST1-expressing breast cancer may be sensitive to anti-PD-L1 immunotherapy.

## 4. Discussion

Multiple EMT-inducing transcription factors, such as TWIST1, SNAIL1/2, and ZEB1/2, may be expressed to drive breast cancer cells to undergo EMT and produce increased invasion, metastasis, and resistance to therapies, including immunotherapy. We have previously shown that the ectopic expression of TWIST1 alone can effectively reprogram the luminal epithelial, non-metastatic ER+/HER2− MCF7 breast cancer cells into mesenchymal, metastatic ER−/HER2− basal-like breast cancer cells. The knockdown of TWIST1 in TNBC cells can significantly inhibit their invasion and metastasis [25,26,27]. Tumor cell-specific knockout of *Twist1* in the mammary gland tumors of a genetically engineered mouse model largely prevented these tumor cells from undergoing EMT, intravasation, and distant metastasis [20]. Liao et al. also showed that IL-17A upregulated EMT-inducing transcription factors including Twist1, N-cadherin and Snail, as well as PD-L1 in NSCLC cells, resulted in increased cell invasion, colony formation ability, tumor growth, and response to anti-PD-L1 therapy [61]. Occurring in parallel to EMT and metastasis are changes in the tumor microenvironment that suppress anti-tumor immunity. Although EMT has been proposed to be associated with cancer immune evasion, reports about the molecular linkages between EMT and the suppression of anti-tumor immunity have been limited. Previous studies have proposed an indirect crosstalk molecular mechanism among EMT, miR-200, ZEB1, PD-L1, and metastasis in cancer cells. In epithelial cells, miR-200 is high, and it targets ZEB1 and PD-L1 mRNAs and suppresses EMT and metastasis. In EMT cancer cells, ZEB1 is high, and it transcriptionally represses miR-200 expression, resulting in increased PD-L1 and enhanced EMT and metastasis [46,48,62,63]. In addition, SNAIL is also reported to indirectly induce PD-L1 expression through activation of the Wnt pathway [64].

In this study, we found that the expression levels of TWIST1 and its direct target genes as well as PD-L1 are the highest in ER−/HER2− breast cancers. High PD-L1 expression is also positively associated with TWIST1 expression in metastatic breast cancers and TNBC cell lines. Furthermore, the expression of TWIST1 in MCF7-TWIST1 cells induced the robust expression of PD-L1, while the knockdown of TWIST1 in BT549 TWIST1-positive cells drastically downregulated PD-L1. These novel observations clearly demonstrate that TWIST1 strongly upregulates PD-L1 expression in ER−/HER2−/TWIST1+ breast cancer cells in addition to inducing EMT to promote breast cancer invasion and metastasis. Therefore, these data allow us to establish a link between TWIST1-induced EMT and an EMT-associated increase in PD-L1 expression for the first time, which supports the notion that TWIST1 may promote the immune escape of breast cancer cells through upregulating PD-L1 expression.

We have recently demonstrated that TWIST1 binds to *SNAI2* and *MYC* genes and recruits TIP60-Com to transcriptionally upregulate these genes [26]. In this study, we found that TWIST1 employs a similar mechanism to upregulate PD-L1. TWIST1 binds to the promoter of the *CD274* gene, recruits TIP60-Com to the promoter through interacting with BRD8 in the TIP60-Com, increases the level of acetylated histone H4, and robustly upregulates *CD274* mRNA and its protein PD-L1. We also showed that the knockdown of endogenous TWIST1 or BRD8 in BT549 TNBC cells drastically decreased PD-L1 expression to an exceedingly low level. These findings indicate that *CD274* is a direct target gene of TWIST1 and that TWIST1 relies on BRD8 to recruit TIP60-Com to transcriptionally promote *CD274* (PD-L1) expression.

The dynamic interactions between immune and cancer cells in the tumor microenvironment play crucial roles in cancer progression. Among the various immune cells, tumor neoantigen-specific CD8+ T cells appear to be key immune cells showing strong cytotoxic activity [65]. An abundance of CD8+ T cells in the tumor microenvironment correlates with improved survival [66], while PD-L1 expression in tumor and other cells in the tumor microenvironment correlates with poor survival rates in breast cancer patients [67]. It is also well-established that high-level PD-L1 on the cancer cell surface binds its receptor PD-1 on the CD8+ T-cell surface, leading to the suppression and exhaustion of CD8+ T cells and facilitating cancer immune evasion. Indeed, the exhaustion of CD8+ T cells has been one of the major barriers to tumor immunity, and reinvigorating those exhausted CD8+ T cells can provide opportunities to improve anti-tumor immunotherapy. In this study, we demonstrated that TWIST1-upregulated PD-L1 in breast cancer cells significantly suppresses the cancer cell-killing activity of CD8+ T cells and accelerates their exhaustion and death in the coculture. We also demonstrated that the blockade of PD-L1 function with a PD-L1-neutralizing antibody or knockdown of TWIST1 to diminish TWIST1-induced PD-L1 expression in breast cancer cells greatly reversed the exhaustion status of CD8+ T cells and reinvigorated CD8+ T cells to inhibit breast cancer cells in the coculture. These results indicate that the TWIST1-induced PD-L1 expression in breast cancer cells greatly contributes to the functional suppression and exhaustion of cytotoxic CD8+ T cells. These results also suggest that the blockade of TWIST1-upregulated PD-L1 function in breast cancer cells may help to prevent the immune escape of breast cancer cells and that TWIST1-expressing breast cancer cells may be sensitive to ICI-based immunotherapies.

The normal mammary gland contains both ER+ and ER− luminal epithelial cells. Our previous cell lineage-tracing study demonstrated that breast cancer cells can originate from both ER+ and ER− luminal epithelial cells. During tumor progression, the cancer cells originating from ER− cells maintain ER− status, while those originating from ER+ cells can either maintain their ER+ and slow-proliferation status or lose their ER expression to become ER− and more aggressive breast cancer cells and gain fast-proliferation status [68]. Therefore, an ER− or TNBC cancer can originate from either an ER+ or ER− mammary epithelial cell. The MCF7-Ctrl and MCF7-TWIST1 cells used in this study represent a progression model from ER+/HER2− to ER−/HER2− breast cancer cells reprogrammed by TWIST1-induced EMT. During this progression, TWIST1 also induced PD-L1 expression to enable the cancer cells to suppress immune clearance. The BT549 cell line is an established TNBC cell mode with endogenous high TWIST1 and PD-L1 expression, although its origin of mammary epithelial cell lineage is unknown. Using this cell model, we demonstrated that TWIST1 is responsible for maintaining high-level PD-L1 expression. Results from both cell models suggest that there is a TWIST1-PD-L1 regulatory axis to drive PD-L1 expression, which is associated with TWIST1-induced EMT. We are aware of the limitations of these experiments, including the usage of only two independent parent cell lines and the requirement of more cell lines, especially more TNBC cell lines, to generalize this regulatory axis to all TWIST1+ breast cancer cells. In future experiments, using a xenograft breast cancer syngeneic mouse model with an intact immune system to test how the alteration of TWIST1 expression and/or anti-PD-L1 therapy will affect tumor growth, CD8+ T-cell exhaustion, and immunotherapy efficacy will further confirm the role of TWIST1 in breast cancer immune evasion. Furthermore, an examination of TWIST1 and PD-L1 protein expression profiles in individual cells of human breast tumor specimens will further validate the regulatory relationship between TWIST1 and PD-L1 as well as their prognosis relevance.

Existing data strongly suggest TWIST1 as a molecular target of metastatic breast cancer as it promotes EMT, cell invasion, metastasis, and drug resistance. Now, this study further suggests that TWIST1 upregulates PD-L1 in breast cancer cells to cause CD8+ T-cell exhaustion. In addition, our previous study in an inducible TWIST1 knockout mouse model also demonstrated that the global knockout of TWIST1 in adult mice does not cause any obvious health problem, although the germline knockout of TWIST1 causes embryonic lethality, suggesting that the inhibition of TWIST1 function in adult patients with cancer would not cause severe adverse effects [69]. These data clearly indicate that TWIST1 should be a molecular target to treat breast cancer. However, TWIST1 is a nuclear protein that is not an easy target. Harmine has been reported to cause TWIST1 degradation [70], but the clinical application of this molecule has not been reported yet. In addition, a recombinant TWIST1 protein has been used as a vaccine to induce both CD8+ and CD4+ TWIST1-specific T-cell responses in vivo to attack TWIST1-expressing breast tumor cells with certain efficacy [71]. We believe that the future development of powerful small molecular inhibitors that directly inhibit TWIST1 transcriptional function will have significant clinical applications in treating metastatic breast cancers with TWIST1 and TWIST1-upregulated PD-L1 expression.

## 5. Conclusions

TWIST1 expression is positively correlated with *CD274*/PD-L1 expression in metastatic breast cancers and TNBC cell lines. *CD274* is a direct target gene of TWIST1, and TWIST1 binds to the *CD274* promoter and recruits TIP60-Com in a BRD8-dependent manner to transcriptionally upregulate *CD274*/PD-L1 expression. The TWIST1-upregulated PD-L1 in breast cancer cells drives immune evasion in vitro by suppressing the cancer cell-killing activity and causing the exhaustion and death of CD8+ T cells. The knockdown of TWIST1 or BRD8 or the blockade of PD-L1 activity can successfully reverse CD8+ T-cell exhaustion and reinvigorate CD8+ T cells, leading them to inhibit the growth of breast cancer cells. Future experiments of xenograft mouse models using syngeneic mouse breast tumor cells with engineered TWIST1 or BRD8 expression in immune-competent mice treated with or without the anti-PD-L1 antibody should help to address whether TWIST1-induced PD-L1 expression truly drives the immune evasion of breast cancer. These findings suggest that TWIST1 and BRD8 may serve as potential targets for preventing PD-L1 overexpression-caused CD8+ T-cell exhaustion and death.

## Figures and Tables

**Figure 1 cancers-16-01973-f001:**
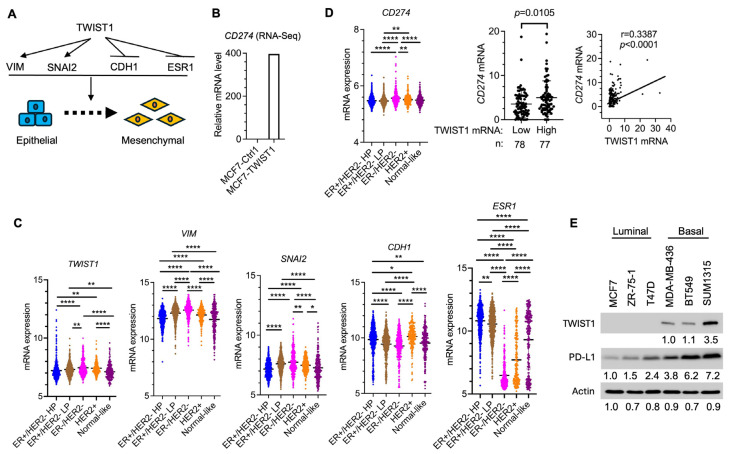
CD274 expression is positively correlated with the expression levels of TWIST1 and its target genes in ER−/HER2− breast cancer. (**A**). TWIST1 upregulates mesenchymal genes such as *VIM* and *SANI2* and represses epithelial genes such as *CDH1* and *ESR1* to induce EMT. (**B**). RNA-Seq data suggested a robust increase in *CD274* mRNA in MCF7-TWIST1 cells versus MCF7-Ctrl1 cells. (**C**). Expression profiles of *TWIST1*, *VIM*, *SNAI2*, *CDH1*, and *ESR1* mRNAs in the indicated five subtypes of human breast tumors, which were obtained from analysis of the METABRIC datasets of 1970 clinical specimens including 617 ER+/HP (high proliferation), 640 ER+/LP (low proliferation), 309 ER−/HER2−, 198 HER2+, and 216 normal-like tumors. The *p*-values were calculated by one-way ANOVA. *, *p* < 0.05; **, *p* < 0.01; ****, *p* < 0.0001. (**D**). Left: The expression profile of *CD274* mRNA in the five subtypes of human breast tumors from analysis of the METABRIC datasets. The *p*-values were calculated as described above. Central: The average expression levels of *CD274* mRNA in TWIST1-low (*n* = 78) and TWIST1-high (*n* = 77) metastatic breast cancer datasets (*n* = 155). The *p*-value was determined by an unpaired *t*-test. Right: The correlation between *TWIST1* and *CD274* mRNA expression levels in the metastatic breast cancer datasets (*n* = 155) was determined by Pearson analysis. (**E**). Western blot analysis (Appendix A: Original Western Blots) of TWIST1 and PD-L1 (CD274) proteins expressed in the indicated breast cancer cell lines. β-actin served as a loading control. The relative band intensity for each protein was normalized to the band intensity of β-actin and presented as a number under each band.

**Figure 2 cancers-16-01973-f002:**
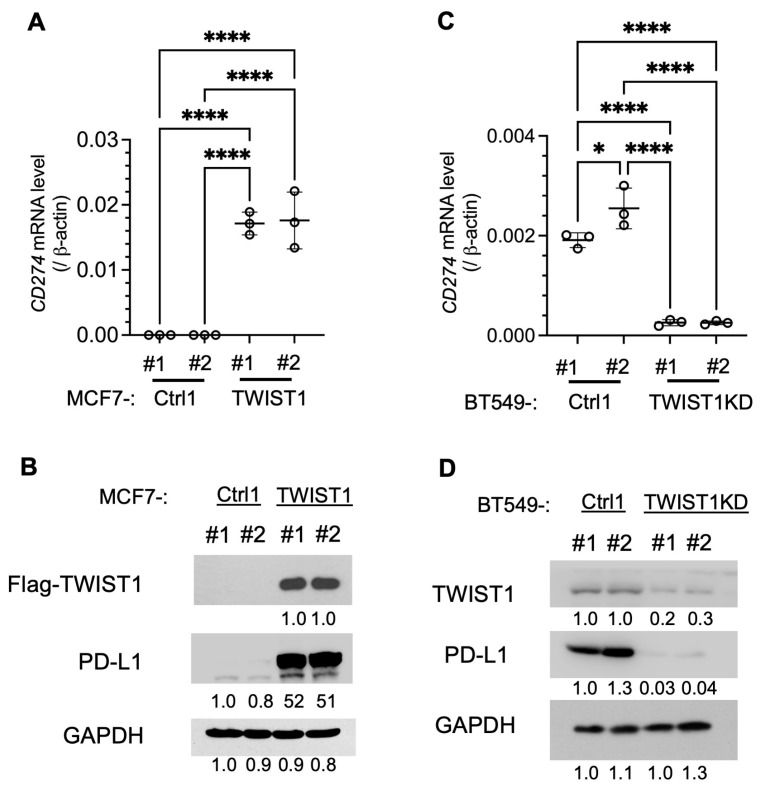
Expression and knockdown of TWIST1 increased and decreased CD274 expression, respectively. (**A**,**B**). CD274 mRNA and its protein PD-L1 expressed in two different lines of MCF7-Ctrl1 and MCF7-TWIST1 cells, which were measured by RT-qPCR and Western blot, respectively. The *p*-values were calculated by one-way ANOVA. ****, *p* < 0.0001. (**C**,**D**). *CD274* mRNA and its protein PD-L1 expressed in two lines of BT549-Ctrl1 and BT549-TWIST1KD cells, which were measured as described above. The *p*-values were calculated as described above. *, *p* < 0.05. The data points in panels (**A**) and (**C**) represent biological replicates of assays.

**Figure 3 cancers-16-01973-f003:**
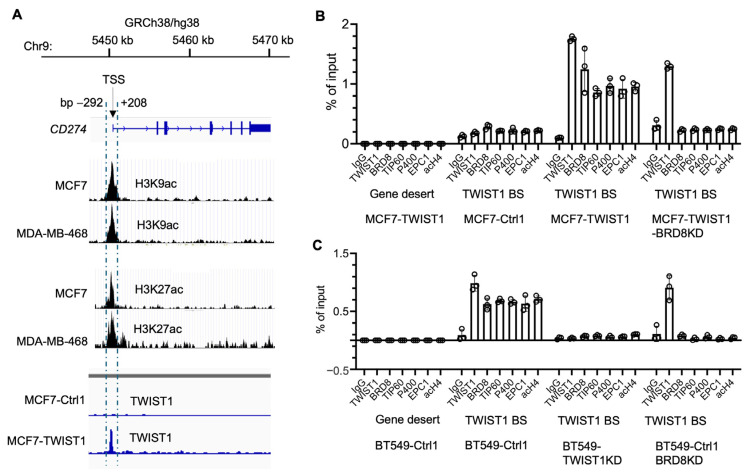
TWIST1 binds the *CD274* promoter and recruits the TIP60 complex in a BRD8-dependent manner. (**A**). The location of the human *CD274* gene in Chromosome 9 is indicated according to UCSC Genome Browser on Human (GRCh38/hg38). The transcriptional start site (TSS) of *CD274* is indicated. The acetylated histone H3K9ac and H3K27ac peaks from bp −292 to +208 to the *CD274* TSS in MCF7 cells [57,58] and MDA-MB-468 cells (ViMIC Database GSM2258896 and GSM2258884) are aligned with the *CD274* gene. The TWIST1 binding peak in the same region in MCF7-TWIST1 cells was mapped by ChIP-Seq as described previously, where MCF7-Ctrl1 cells without TWIST1 expression served as a negative control for the ChIP-Seq assay [26]. (**B**,**C**). ChIP-qPCR assays for a gene desert negative control region and the TWIST1-binding site (TWIST1 BS) that overlaps with the H3K9ac and H3K27ac peaks near the *CD274* TSS (see panel (**A**)). ChIP-qPCR assays were performed using the indicated cell lines and the antibodies against the indicated proteins, and the non-immune IgG as a negative control. Quantitative data for each bar were obtained from three biological replicate assays.

**Figure 4 cancers-16-01973-f004:**
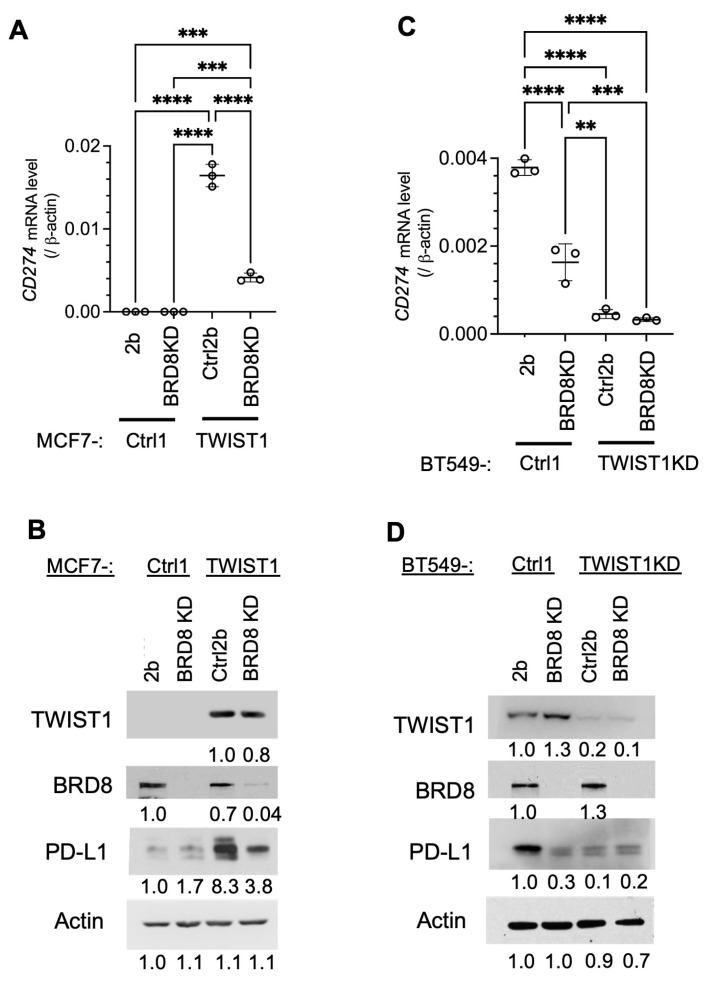
Knockdown of BRD8-diminished TWIST1-promoted expression of CD274 mRNA and protein. (**A**,**B**). CD274 mRNA and its protein PD-L1 were measured in MCF7-Ctrl1-2b, MCF7-Ctrl1-BRD8KD, MCF7-TWIST1-Ctrl2b, and MCF7-TWIST1-BRD8KD cells by RT-qPCR and Western blot. The *p*-values were calculated by one-way ANOVA. ***, *p* < 0.001; ****, *p* < 0.0001. (**C**,**D**). CD274 mRNA and its protein PD-L1 were measured in BT549-Ctrl1-2b, BT549-Ctrl1-BRD8KD, BT549-TWIST1KD-Ctrl2b, and BT549-TWIST1KD-BRD8KD cells by RT-qPCR and Western blot. The *p*-values were calculated as described above. **, *p* < 0.05. The data points in panels (**A**) and (**C**) represent biological replicates of assays.

**Figure 5 cancers-16-01973-f005:**
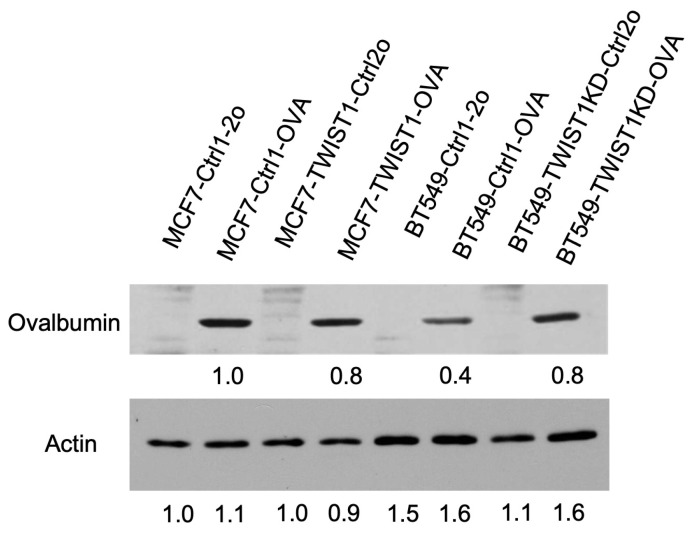
Generation of stable OVA-expressing cell lines. The cell lines established previously were transduced with lentiviral particles that express only mCherry (labeled by 2o or Ctrl2o in the cell line names), or both mCherry and Ovalbumin (labeled by OVA in the cell line names). The stabilized mCherry-expressing cells were sorted out by flow cytometry and expanded. Western blot was performed using an OVA-specific antibody. β-actin served as a loading control.

**Figure 6 cancers-16-01973-f006:**
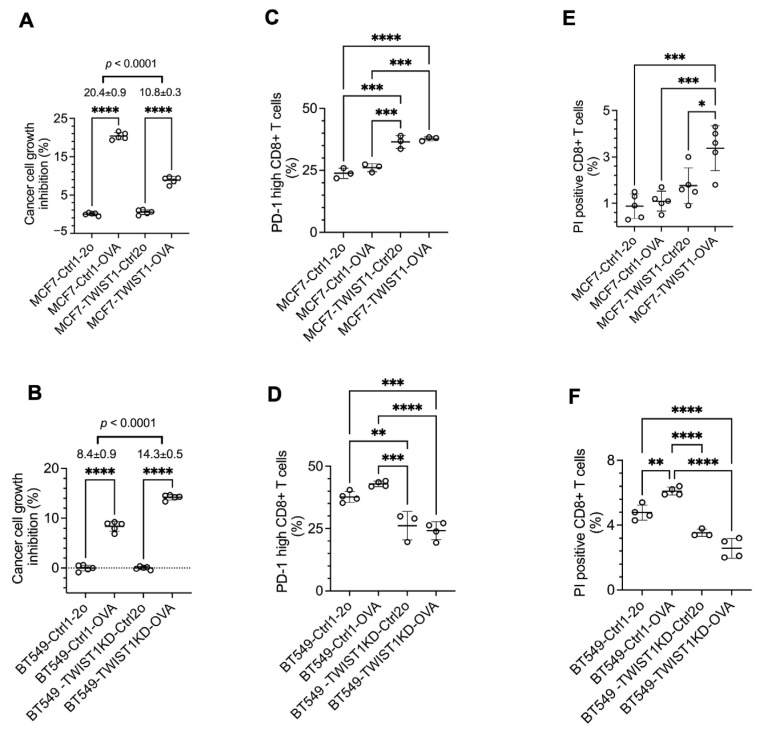
TWIST1 promotes immune escape by inhibiting CD8+ T cells in vitro. (**A**,**B**). The indicated breast cancer cells with or without OVA expression were cocultured with OT-1 CD8+ T cells at a 1:4 ratio for 3 days. The viability of the adherent breast cancer cells in each coculture was measured by MTS assay. The relative percentage of cell growth inhibition for each OVA-expressing cell line was normalized by setting the relative percentage of cell growth inhibition for its matched control cell line at 0%. The *p* values were calculated by unpaired *t*-test. ****, *p* < 0.0001. (**C**,**D**). OT-1 CD8+ T cells were cocultured with the indicated breast cancer cell lines with or without OVA expression at the same ratio for the same time period as described in panels A and B. The percentages of PD-1 (CD279)-high OT-1 CD8+ T cells out of total OT-1 CD8+ T cells were measured by flow cytometry. The *p*-values were calculated by one-way ANOVA. *, *p* < 0.05; **, *p* < 0.01; ***, *p* < 0.001; ****, *p* < 0.0001. (**E**,**F**). OT-1 CD8+ T cells were cocultured with the indicated breast cancer cell lines with or without OVA expression at the same ratio for the same time period as described in panels (**A**,**B**). PI staining was applied to the total unfixed OT-1 CD8+ T cells collected from the coculture to detect PI-permeable apoptotic cells. The percentages of PI staining-positive OT-1 CD8+ T cells out of total OT-1 CD8+ T cells were measured by flow cytometry. The *p* values were determined as described above. The data points in all panels represent biological replicates of assays.

**Figure 7 cancers-16-01973-f007:**
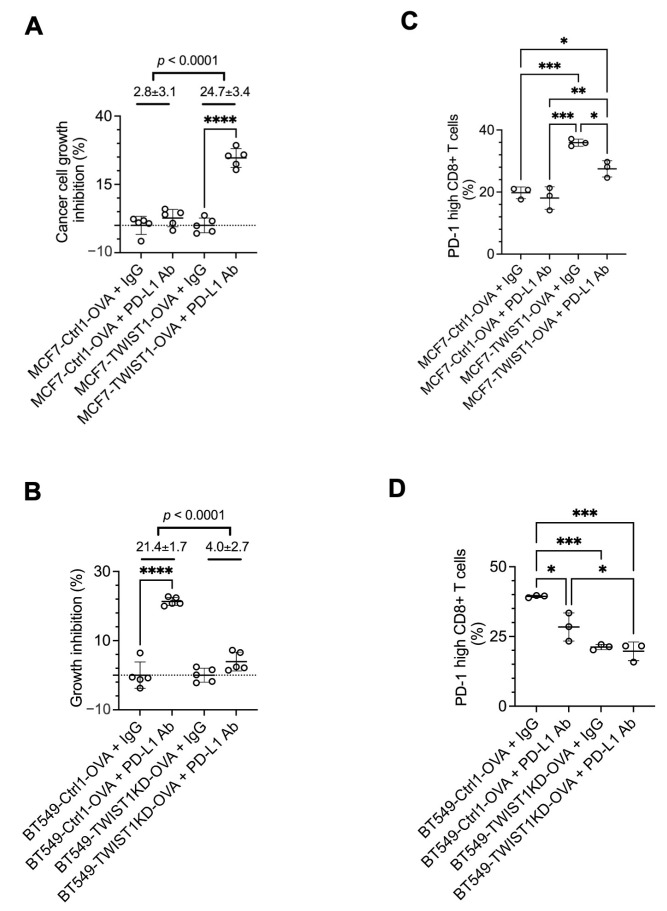
Inhibition of PD-L1 with an antibody strongly enhanced CD8+ T cells to kill breast cancer cells with TWIST1 expression. (**A**,**B**). The indicated breast cancer cells with OVA expression were cocultured with OT-1 CD8+ T cells at a 1:4 ratio and treated with non-immune IgG or PD-L1 antibody for 3 days. The viability of the adherent breast cancer cells in each coculture was measured by MTS assay. The relative percentage of cell growth inhibition for each cell line treated with PD-L1 antibody was normalized by setting the relative percentage of cell growth inhibition for the same cell line treated with IgG at 0%. The *p* values were calculated by unpaired *t*-test. ****, *p* < 0.0001. (**C**,**D**). OT-1 CD8+ T cells were cocultured with the indicated breast cancer cell lines with OVA expression and treated with non-immune IgG or PD-L1 antibody as described in panels (**A**,**B**). The percentages of PD-1-high OT-1 CD8+ T cells out of total OT-1 CD8+ T cells were measured by flow cytometry. *p*-values were calculated by one-way ANOVA. *, *p* < 0.05; **, *p* < 0.01; ***, *p* < 0.001. The data points in all panels represent biological replicates of assays.

**Table 1 cancers-16-01973-t001:** The nomenclature and relationships of cell lines derived from MCF7 and BT549 cells.

First Generation	Second Generation	Third Generation	
		Cell Lines	Selection Marker(s)
MCF7	MCF7-Ctrl1 (G418)	MCF7-Ctrl1-2b	G418 and puromycin
		MCF7-Ctrl1-BRD8KD	
		MCF7-Ctrl1-2o	G418 and mCherry
		MCF7-Ctrl1-OVA	
	MCF7-TWIST1 (G418)	MCF7-TWIST1-Ctrl2b	G418 and puromycin
		MCF7-TWIST1-BRD8KD	
		MCF7-TWIST1-Ctrl2o	G418 and mCherry
		MCF7-TWIST1-OVA	
BT549	BT549-Ctrl1 (puromycin)	BT549-Ctrl1-2b	puromycin
		BT549-Ctrl1-BRD8KD	
		BT549-Ctrl1-2o	puromycin and mCherry
		BT549-Ctrl1-OVA	
	BT549 TWIST1KD (puromycin)	BT549-TWIST1KD-Ctrl2b	puromycin
		BT549-TWIST1KD-BRD8KD	
		BT549-TWIST1KD-Ctrl2o	puromycin and mCherry
		BT549-TWIST1KD-OVA	

## Data Availability

Data were generated by the authors and are included in the article.

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
