# Peer review of "TWIST1 Drives Cytotoxic CD8+ T-Cell Exhaustion through Transcriptional Activation of CD274 (PD-L1) Expression in Breast Cancer Cells"

_cancers, 2024, doi:10.3390/cancers16111973_

Round 1

Reviewer 1 Report

Comments and Suggestions for Authors

This manuscript titled as "TWIST1 Drives Immune Evasion through Upregulating PD-L1 Expression in Breast Cancer" tries to address a topic in breast cancer research by exploring the role of TWIST1 in regulating PD-L1 expression and its implications for immune evasion. However, apparent flaws exist in this study which requires substantial revision. The design of this study is seriously disconnected from clinical practice and lacks clinical significance, thus this study is recommended to be improved by an oncology physician who knows well about BS cancer internal therapy. Main issues are listed below:

1.Authors should clarify the subtypes of BC in the title of this manuscript based on the outcomes of this study instead of using “Breast Cancer”.

2.This study is seriously disconnected from clinical practice and lacks clinical significance. As far as I know, TNBC is currently only subtype of BC that have been officially approved to be treated by PD-1/PD-L1 inhibitors, which means there is not enough evidence to support the efficiency of regulating PD-L1 in other subtypes of BC. Thus, it was illogical and pointless that MCF7 and BT549 had been selected for this study. 

3.Since this manuscript concludes that “These results suggest that TWIST1 may serve as a target to control immune evasion and a biomarker to predict the value of immune checkpoint inhibitor therapy” and “TWIST1 may also serve as a biomarker for anti-PD-L1 immunotherapy”, the effects of regulating TWIST1 on the therapeutic effect of PD-1/PD-L1 inhibition therapy on treating TNBC should be evaluated for improving the clinical significance of this study.

4.At least 3 different TNBC cells should be added in this study to confirm authors’ hypothesis.

5.In vivo study is strongly required.

6.A more thorough exploration of potential confounders, the limits of the experimental models used, and the implications of these limitations for the study’s conclusions are recommended.

7.Elaboration on why TWIST1 is a particularly important target for study beyond general statements about EMT and metastasis is recommended. Include more specific prior findings linking TWIST1 to immune responses if available.

8.A clearer organization of sections and more concise language are recommended. Some sentences are overly complex, making them difficult to follow.

9.Detailed information about the statistical methodologies of this study should be added. It should include more specifics about the analyses, such as the exact statistical tests used and justification for their use, to ensure transparency and replicability.

10.The discussion could be expanded to better situate the findings within the larger body of literature on EMT, PD-L1 expression, and immune evasion in cancer. Comparative analysis with other EMT-inducing transcription factors could enrich the discussion.

11.Authors should provide more details about the controls used in each experiment, especially in cell culture experiments and Western blots, to help clarify how conclusions were drawn.

12.Potential clinical applications of the research should be added. How might these findings influence current treatment strategies for breast cancer? What are the next steps in research to move from these findings toward clinical application?

Author Response

Please see uploaded pdf file for point-by-point responses to Reviewer's comments. 

Reviewer 2 Report

Comments and Suggestions for Authors

The study by Yu et al. provides insights into how certain proteins, Twist1 and PD-L1, are involved in cancer. However, there are some concerns that need addressing:

1. The authors should connect their findings with what we know from real human breast cancer data, not just cell line studies. There's a wealth of published datasets (single cell and bulk RNA seq) out there that could help validate their findings with actual patient samples.

2. It would be helpful to understand how reducing Twist1 affects other immune-related markers and pathways. This would give us a clearer picture of how Twist1 influences the immune response in cancer.

3. It's crucial to confirm whether Twist1 truly affects the transition of breast cancer cells into a more aggressive state. This would solidify the link between Twist1 and cancer progression.

4. I suggest that the authors conduct experiments to look at immune cells and Twist1 levels in actual tumor samples using a method called immunohistochemistry. This would help us understand how Twist1 and immune cells interact in the tumor environment.

5. The claims made about immune evasion in the paper might be a bit overstated because there haven't been any experiments done in living organisms (in vivo experiments). Without this type of evidence, we should be careful about drawing conclusions about how cancer cells evade the immune system.

6. Lastly, to really prove their ideas and make them relevant for potential treatments, it's important for the authors to do experiments in living organisms, like mice. This would give us a clearer understanding of how Twist1 and PD-L1 work in a more realistic setting.

Author Response

Please see uploaded document for point-by-point responses to reviewer's comments. 

Reviewer 3 Report

Comments and Suggestions for Authors

Authors: Xiaobin Yu , Jianming Xu  

Title: TWIST1 Drives Immune Evasion through Upregulating PD-L1 Expression in Breast Cancer  

COMMENTS: 

The Authors have performed an interesting study. They have revealed the causal links between the expression of TWIST1, PD-L1 and mechanism of immune evasion in human breast cancer. Their findings may have a significance in theranoistics of breast cancer and be interesting for readers of the journal. The submitted manuscript is well written and illustrated. I think that this manuscript may be accepted in the present form.   

Author Response

Please see the attached document for point-by-point responses to reviewer's comments. 

Round 2

Reviewer 1 Report

Comments and Suggestions for Authors

This manuscript discusses the role of TWIST1 in driving immune evasion in ERα-negative breast cancer through upregulation of PD-L1. Novelty do exist in this article. Additionally, in my previous review, I expressed concern that "This study seems disconnected from clinical practice and lacks clinical significance. As far as I know, the selection of MCF7 and BT549 cell lines for this study seemed illogical and pointless." In their response, the authors clarified that this is a basic research study, which is an acceptable justification. After the authors revised the title of the article as recommended, many parts of the article have become much smoother, which is great. My main concern now is that the study still lacks validation through animal experiments, given that this research is closely related to the in vivo microenvironment. Thus, acceptance is recommended if in vivo confirmation could be added.

Author Response

We are truly grateful to Reviewer 1 for critical evaluating our revised manuscript and accepting most of our revisions, which has greatly helped us to improve the quality of the paper. To do the suggested in vivo experiments, syngeneic mouse models with mouse breast cancer cell lines such as 4T1 or E0071 and matched strain background mice Balb/c or C57BL/6j female mice are required. It will take a long time and a lot of resources to prepare stable cell lines with engineered TWIST1 or BRD8 expression profiles and complete the mouse model experiments. Given these justifications and the limited timeline for publication of this special Issue of Cancers, we decide not to do these experiments for this paper. Again, the novel conceptual findings in this paper are to show that TWIST1 induces robust PD-L1 expression in breast cancer cells, while it induces EMT, invasion and metastasis of breast cancer cells and how TWIST1 uses the TIP60 coactivator complex to transcriptionally active PD-L1 expression. If the editors decide not to accept this manuscript without in vivo data, it is OK for us to withdraw it and submit it to somewhere else for publication.

Reviewer 2 Report

Comments and Suggestions for Authors

Authors should mention future in vivo directions in the conclusion

Author Response

Thanks for the suggestion. We have added the following sentence to the Conclusion: “Future experiments of xenograft mouse models using syngeneic mouse breast tumor cells with engineered TWIST1 or BRD8 expression in immune-competent mice treated with or without anti-PD-L1 antibody should help to address whether TWIST1-induced PD-L1 expression truly drive immune evasion of breast cancer.”

Round 3

Reviewer 1 Report

Comments and Suggestions for Authors

This article is a study on anti-tumor immunity, thus evidence from in vivo studies is crucial. If animal experiments cannot be conducted, I am afraid that the current level of evidence for this study will be difficult to match with the journal of  Cancers.